# GRADIENT-AS-RETRIEVAL: CLASSIFICATION BEYOND CROSS ENTROPY

## ABSTRACT

Cross entropy (CE) is the loss of choice for classification tasks. However, computing the CE loss and gradient requires transcendental functions which may be expensive in emerging computational paradigms such as fully homomorphic encryption for privacy-preserving applications. The transcendental function-free familywise (FW) loss has been shown to enjoy strictly better statistical guarantees than the CE loss. In this work, we prove theoretical results that enable efficient computation of the gradient of the FW loss using "retrieval-style" algorithms. Based on our theory, we provide practical implementations. A challenge in designing new loss functions is that widely adopted optimizers and learning rate schedules are tuned to CE. Experimentally, we demonstrate that the FW loss outperforms cross entropy when we opt for parameter-free learning methods.

## 1 INTRODUCTION

Loss function is a critical component of deep learning models. For instance, in training neural networks, back-propagation starts from taking the gradient of the loss function. Therefore, the gradient of the loss is present in the gradient updates for all the model parameters. Yet compared to other architectural components, the choice of cross entropy has largely remained unchanged in classification tasks, notably in next-token prediction in large language models (LLMs).

The cross entropy is deeply rooted in statistics, appeared as a key ingredient of the negative log likelihood in multinomial logistic regression (Hastie et al., 2009). Moreover, cross entropy enjoys the desirable classification-calibration property (Tewari and Bartlett, 2007; Zhang, 2004).

However, cross entropy is *not* the only loss with this property. Recently, Duchi et al. (2018) shows that the FW loss is superior to the cross entropy from a classification-calibration point of view. From the practical side, Fathony et al. (2016) shows that the familywise loss is effective for training linear classifiers.

However, Duchi et al. (2018) leaves open the questions of (I) computationally efficient implementation of the familywise loss in an automatic differentiation framework and (II) empirical demonstration that the familywise loss actually works in *deep learning* classifiers. We fill both gaps in this work.

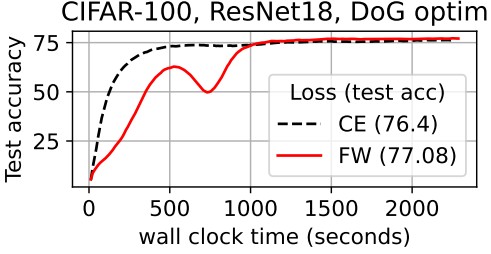

Figure 1: Test accuracy versus wall clock time on training ResNet18 using the DoG optimizer (Ivgi et al., 2023). See Appendix C.3 for additional independent runs with similar outcome.

For $K$-class classification, the familywise loss is defined as

$$\mathcal{L}(\mathbf{z}, y) = 1 - z_y + \max_{\kappa=1,\ldots,K} \left\{ \tfrac{1}{\kappa}\left(\textstyle\sum_{j=1}^{\kappa} z_{(j)}\right) - \tfrac{1}{\kappa} \right\}, \quad \text{where} \quad z_{(1)} \geq z_{(2)} \geq \cdots \geq z_{(K)}. \quad (1)$$

Here, the $\mathbf{z} = (z_j)_{j=1}^{K}$ are the (pre-softmax) logits[1] of a neural network. Unlike the cross entropy, note that (1) does not depend on any transcendental function. Instead, the main computational work

---

[1]The terminology "logit" is typically used only in conjunction with the cross entropy. However, due to the prevalence of the term, we abuse notation and use "logits" when working with family wise loss as well.

is the sorting operation. If we let $\bar{\kappa}$ be the index that attains the max in Equation (1), then we need only the top $\bar{\kappa}$ entries $z_{(1)}, \ldots, z_{(\bar{\kappa})}$ and $z_y$, which affect the gradient of $\mathcal{L}(\mathbf{z}, y)$. In fact, this gradient can be calculated as

$$\nabla_{\mathbf{z}} \mathcal{L}(\mathbf{z}, y) = -\text{one-hot@}y \; + \; (1/\bar{\kappa}) \cdot \text{many-hot@}\{(1), \ldots, (\bar{\kappa})\}. \tag{2}$$

See Remark 2 below for mathematical details. Thus, the majority of computational work is the top-$\bar{\kappa}$ operation on the logits $\mathbf{z}$. It is for this reason that we use the phrase *gradient-as-retrieval* to describe the familywise loss: both the loss and gradient calculations involve only retrieval-like algorithms, and do not involve any transcendental functions.

**Our contributions**. We advance both the theory and application of the familywise along the following directions:

1. **Theory enabling fast gradient computation**. We show that a subgradient of the familywise loss is computable in $\mathcal{O}(K)$ time, where $K$ is the number of classes. To achieve this, we prove a *bitonicity property* (Proposition 3) that enables computing the term inside the "max" in faster-than-sorting (sorting-based is $\mathcal{O}(K \log K)$-time) computation that relies on selection-type algorithms (Theorem 4).

2. **Insights on sparsity**. We uncover and exploit a fundamental difference between the CE and the FW losses. The CE loss gradient is *dense*. By contrast, the gradient of FW loss becomes *sparse* as training proceeds (Figure 6). In view of the gradient formula Equation (2), the sparsity is precisely $\bar{\kappa}/K$. We prove an approximation of $\bar{\kappa}$ the in terms of the (easy-to-compute) variance of logits Theorem 5. This leads to an interpretation of the variance of logits as a form of confidence. See Figure 6.

3. **Efficient implementations**. Leveraging our theory, we implement fast custom kernel for computing the loss and gradient (i.e., the forward and backward) for multicore CPU and pure PyTorch GPU. Our CPU implementation calculates the gradient of the FW loss faster than that of the CE loss when $K$ is large (Figure 2 ). Our pure PyTorch GPU implementation is slower than the CE, but provides speed up on the GPU over naive FW loss implementation (Figure 4). Both implementations on the CPU and GPU exploits theoretical properties we establish.

4. **Experimental results on the FW loss**. We demonstrate the efficacy of the FW loss on real world classification tasks when combined with three different automatically learned learning-rate schedulers: DoG (Ivgi et al., 2023), GDTUO (Chandra et al., 2022) and Prodigy (Mishchenko and Defazio, 2024). See Section 5.

## 2 RELATED WORK

**Previous work on the FW loss**. While less widely known, the FW loss appeared in (Bartlett and Wegkamp, 2008, §2), however only in the binary classification case. Later, Duchi et al. (2018) defined this loss for multiclass classification where $K \geq 2$ and proved the *universal equivalence to 01 loss* property underlying the advantage of FW loss over the CE loss. Below, we will say "universal equivalence" to mean "universal equivalence to 01 loss".

Roughly speaking, the universal equivalence (Duchi et al., 2018, Definition. 4.1) says the following: Let $\mathcal{M}$ be the set of all measurable functions on the sample space, and $\mathcal{Q}$ be either $\mathcal{M}$ itself or a finite subset of $\mathcal{M}$. Let $\{f_n\}_{n=1,2,\ldots}$ be a sequence of functions in $\mathcal{Q}$. Whenever the testing $\mathcal{L}$-risk converges to the Bayes-risk (i.e., the best theoretically achievable $\mathcal{L}$-risk) $\mathbb{E}_{\mathbf{X},Y}[\mathcal{L}(f_n(\mathbf{X}), Y)] \to \inf_{f \in \mathcal{Q}} \mathbb{E}_{\mathbf{X},Y}[\mathcal{L}(f(\mathbf{X}), Y)]$ as $n \to \infty$ the testing *01 loss also converges* to the theoretical best-case scenario $\mathbb{E}_{\mathbf{X},Y}[\text{argmax}(f_n(\mathbf{X})) \neq Y] \to \inf_{f \in \mathcal{Q}} \mathbb{E}_{\mathbf{X},Y}[\text{argmax}(f_n(\mathbf{X})) \neq Y]$ as $n \to \infty$.

This is in contrast to a more widely known definition of *classification-calibration*, which only applies when $\mathcal{Q} = \mathcal{M}$. Thus, universal equivalence is more applicable than classification-calibration. The fact that the FW loss has the universal equivalence property was proved in Duchi et al. (2018, Examples 7 & 8). On the other hand, the CE loss has only the classification-calibration property.

In summary, the FW loss *should*, in theory, outperform the CE loss. However, this does not necessarily mean that FW loss is computationally efficient, nor that theoretical advantage carries over to real world tasks. In this work, we fill both gaps.

**Classification-calibration**. Establishing the desirable property of classificaiton-calibration for binary loss functions was initiated in Bartlett et al. (2006) and refined subsequently by many authors. See Frongillo and Waggoner (2021) and the references there-in. On the multiclass classification side, the work was initiated by Tewari and Bartlett (2007); Zhang (2004).

The theory of classification-calibration played an important role in designing and comparing multiclass loss functions in the context of multiclass linear models. See Fathony et al. (2016); Dogan et al. (2016) for recent works on the impact of the theory on practice.

However, no work has empirically examined the FW loss, which is the only known loss with the universal equivalence property, to the best of our knowledge. Our work fills this gap in the experimental validation supporting the use of FW loss for real world tasks.

**Alternatives and modification to the cross entropy**. Recently, works by Hui and Belkin (2020) and Hui et al. (2023) proposed the square loss and a square loss-based modification to the cross entropy known as "squentropy", based on the empirical performance. Another popular line of works is in augmenting the CE loss for robustness to label noise (Zhang and Sabuncu, 2018; Amid et al., 2019).

As pointed by Hui et al. (2023), existing training "recipes" for classification tasks are based on the CE loss. They conjecture that gains can be made for alternatives to the CE loss if one adapts more approaches agnostic to loss functions. We confirm that this is the case for the FW loss in our experiments. When we use an off-the-shelf auto learning rate schedulers (Chandra et al., 2022; Ivgi et al., 2023; Mishchenko and Defazio, 2024), we see that using the FW loss outperforms the CE loss. See Section 5 below.

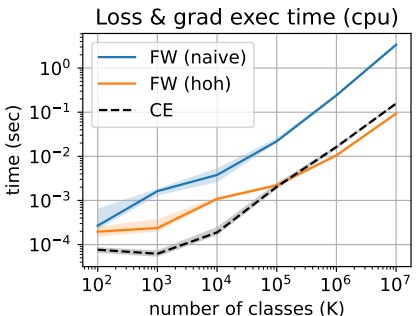

**Transcendental functions and "expensive" loss functions**. In earlier works on training linear models, (Yuan et al., 2011) showed that transcendental functions such as $\exp$ and $\log$ were expensive bottlenecks. This issue is alleviated on GPUs with fast special function units (SFUs) dedicated to computing $\exp$ and $\log$. But the issue has re-emerged in the context of privacy-preserving machine learning.

Figure 2: A loss-and-grad calculation speed comparison between the naive implementation of the familywise loss (Appendix D), heap-of-heaps based implementation, and PyTorch's cross entropy. Each run is repeated 20 times. The 20-80 quantiles are plotted as fill-betweens.

Cho et al. (2024) highlights the difficulty of computing softmax in the framework of fully homomorphic encryption (FHE). For privacy-preserving machine learning, FHE is being explored as a key component and allows arithmetic computation in encrypted ciphertext. However, transcendental functions remain challenging. For this reason, we believe investigating transcendental function-free architecture, as we have done in this work, is an important future direction.

**Retrieval-style algorithms for CPUs and GPUs**. It is true that our loss computation is superior to the CE loss only on the CPU. However, we hope our work will inspire more researchers from the "GPU for database" and parallel algorithms community to consider machine learning as another application area (Shanbhag et al., 2018; Kuszmaul and Westover, 2020).

## 3 FAMILYWISE LOSS

The goal of this section is to establish the theoretical foundation for efficient computation of the familywise loss and gradient. The complete implementations will appear in Section 4 below. For reader's convenience, we restate the definition of the familywise loss which was introduced in Example 3 of Duchi et al. (2018):

**Definition 1.** *Let* $\mathbf{z} \in \mathbb{R}^K$ *and* $y \in \{1, \ldots, K\}$. *The* familywise loss *is defined by*

$$\mathcal{L}(\mathbf{z}, y) = 1 - z_y + \max_{\kappa \in \{1, \ldots, K\}} \left\{ -\frac{1}{\kappa} + \frac{1}{\kappa} \sum_{j=1}^{\kappa} z_{(j)} \right\} \tag{3}$$

*where* $z_{(1)} \geq z_{(2)} \geq \cdots \geq z_{(K)}$ *is the entries of* $\mathbf{z}$ *sorted in descending order. Let* $\bar{\kappa}(\mathbf{z}) \in \{1, \ldots, K\}$ *be the index that attains the maximum in* (3). *Throughout, whenever* $\mathbf{z}$ *is unambiguous, we simply write* $\bar{\kappa}$ *instead of* $\bar{\kappa}(\mathbf{z})$.

Below, we will refer to the expression inside the "max" in Equation (3), i.e.,

$$H_\kappa := -\frac{1}{\kappa} + \frac{1}{\kappa} \sum_{j=1}^{\kappa} z_{(j)} \tag{4}$$

as the $\kappa$-th *sorted harmonically-normalized cumulative sum* (abbrev. SHANCS).

**Remark 2.** *Suppose that $\bar{\kappa}$ attains the maximum in* (3)*, then the familywise loss reduces to*

$$\mathcal{L}(\mathbf{z}, y) = 1 - z_y + -\frac{1}{\bar{\kappa}} + \frac{1}{\bar{\kappa}} \sum_{j=1}^{\bar{\kappa}} z_{(j)}. \tag{5}$$

*Taking the gradient with respect to $\mathbf{z}$, we get the formula* (2) *presented earlier in the introduction:*

$$\nabla_{\mathbf{z}} \mathcal{L}(\mathbf{z}, y) = -\text{one-hot}@y + (1/\bar{\kappa}) \cdot \text{many-hot}@\{(1), \ldots, (\bar{\kappa})\} \tag{6}$$

*where one-hot$@y$ (resp. many-hot$@\{(1), \ldots, (\bar{\kappa})\}$) denotes the $K$-dimensional vector with "1" at index $y$ (resp. $j$ for each $j \in \{(1), \ldots, (\bar{\kappa})\}$).*

Given $\bar{\kappa}$ and the value $z_{(\bar{\kappa})}$, note that it is easy to compute both the gradient and the loss. It is not even necessary to sort the top-$\bar{\kappa}$ subvector, since both Equation (6) and Equation (3) only requires the unordered set of top-$\bar{\kappa}$ indices and logit values. However, computing Equation (3) ostensibly requires computing the $\kappa$-th SHANCS for every $\kappa$, which would in turn require fully sorting the entire vector $\mathbf{z}$. We call this the "naive" implementation and include the PyTorch code in Appendix D using the autograd-enabled `torch.sort` operator.

When $K$ becomes large, the $\mathcal{O}(K \log K)$ sorting operation becomes a computational bottleneck. Therefore, a natural question is whether the sorting operation can be substituted by computationally lighter "retrieval-style" algorithms such as the top-n selection (Shanbhag et al., 2018; Ribizel and Anzt, 2020), or partition (Kuszmaul and Westover, 2020).

## 3.1 BITONICITY OF SHANCS

In this section, we establish a "bitonicity" property (Proposition 3) that is key to fast computation of the familywise loss using retrieval-style algorithms. More precisely, we prove that the sequence of $\kappa$-th SHANCS is bitonic in the index $\kappa$ in the following sense: $\kappa$-th SHANCS is first non-decreasing then becomes non-increasing with respect to $\kappa$. Therefore, $\bar{\kappa}$ is precisely an index where the $\kappa$-th SHANCS switches from being non-decreasing to non-increasing.

**Proposition 3** (Bitonicity). *Let $z_1, \ldots, z_K \in \mathbb{R}$ be arbitrary numbers and $z_{(1)} \geq \cdots \geq z_{(K)}$ be these numbers sorted in descending order. Define $H_\kappa := -\frac{1}{\kappa} + \frac{1}{\kappa} \left( \sum_{j=1}^{\kappa} z_{(j)} \right)$. Then $H_1, H_2, \ldots$ is unimodal, i.e., there exists $\bar{\kappa} \in \{1, \ldots, K\}$ such that*

$$H_1 \leq \cdots \leq H_{\bar{\kappa}-1} \leq H_{\bar{\kappa}} \geq H_{\bar{\kappa}+1} \geq \cdots \geq H_K.$$

For the proof of Proposition 3, see Appendix A.1 in the supplementary material. The consequence of this result is the construction of an $\mathcal{O}(K)$ time algorithm for computing $\bar{\kappa}$.

**Theorem 4.** *Let $\mathbf{z} \in \mathbb{R}^K$ and $y \in \{1, \ldots, K\}$ be arbitrary. Let $\mathcal{L}$ be the familywise loss. The gradient $\nabla_{\mathbf{z}} \mathcal{L}(\mathbf{z}, y)$ with respect to $\mathbf{z}$ can be computed in $\mathcal{O}(K)$ time.*

The full proof will be given in Appendix A.2. Below, we present a sketch:

*Sketch of proof of Theorem 4.* The high-level idea is to use the binary search to calculate $H_\kappa$ for specific $\kappa$'s. Below, we sketch the first iteration of the binary search. Instead of fully sorting $\mathbf{z}$, we use a "partition in the middle" algorithm to rearrange $\mathbf{z}$ as follows: Denote by

$$\mathbf{z}' = (z_1', z_2', \ldots, z_K')$$

the rearranged version of $\mathbf{z}$. The every element in the first half of $\mathbf{z}'$ are greater than or equal to the second half of $\mathbf{z}'$, i.e.,

$$z_{j_1}' \geq z_{j_2}' \quad \text{for each } j_1 = 1, \ldots, \lfloor K/2 \rfloor \text{ and } j_2 = \lfloor K/2 \rfloor + 1, \ldots, K.$$

From this, we can compute $H_{\lfloor K/2 \rfloor}$ and $H_{\lfloor K/2 \rfloor+1}$ in $\mathcal{O}(K)$ time.

By checking if $H_{\lfloor K/2 \rfloor} \leq H_{\lfloor K/2 \rfloor + 1}$ and invoking Proposition 3, we can decide to proceed left or right in the binary search. If $H_{\lfloor K/2 \rfloor} \leq H_{\lfloor K/2 \rfloor + 1}$, then we know that $\bar{\kappa} \geq \lfloor K/2 \rfloor$ in which case we repeat the procedure described above on the right half subvector of $\mathbf{z}'$. The total time complexity is thus $\mathcal{O}(K + K/2 + K/4 + K/8 + \cdots) = \mathcal{O}(K)$ □

The algorithm in Theorem 4 uses an average case linear time algorithm for partition which is efficient in theory but not in practice. See references in Alexandrescu (2016). For this reason, our implementations in Section 4 leverage Proposition 3 in a different way. Before moving on to discussing the implemenations, we introduce another trick: a heuristic for approximating $\bar{\kappa}$ based on easy-to-compute quantities.

### 3.2 Approximating $\bar{\kappa}$ under Gaussianity assumption on logits

In this subsection, we estimate $\bar{\kappa}$ under a Gaussian distributional assumption on the logits $\mathbf{z}$.

**Theorem 5.** *Suppose that the logits $\mathbf{z} \in \mathbb{R}$ is distributed according to the distribution $\mathcal{N}(\mu, \sigma/K)$ where $\sigma$ is assumed to be known. Then, as $K \to \infty$, $\bar{\kappa}$ can be approximated as $(1/\sigma)\sqrt{2\log(\sigma)}$.*

We give the full proof of Theorem 5 in Appendix B and give a sketch below:

*Proof.* When we generate samples $\mathbf{z} \sim \mathcal{N}(0, \sigma/K)$ and plot the points $\left(\frac{\kappa}{K}, H_\kappa\right)$ for each $\kappa = 1, \ldots, K$, a limiting curve emerges (Figure 3-left panel). We show that this curve is equal to the function $(0, 1) \ni x \mapsto \sigma \mathrm{ES}(x) - 1/x$ where ES is the so-called expected shortfall of the Gaussian distribution (Norton et al., 2021). Thus, to calculate $\bar{\kappa}$ as $K \to \infty$, it suffices to find the maximizer of $\sigma \mathrm{ES}(x) - 1/x$ over $x \in (0, 1)$.

We show that the function is concave and thus it suffices to find a root of the derivative. We approximate the root of the derivative, resulting in the formula in Theorem 5. □

The approximation in Theorem 5 holds for when $K$ is large. However it is already quite accurate when $K = 10,000$ (Figure 3). When $K = 100$ as in CIFAR-100, we see that the approximation is still reasonable (Figure 6).

**Remark 6.** *In practice when $\sigma$ is not known, we can apply the plug-in method and substitute an empirical estimator $\hat{\sigma}$ into Theorem 5, e.g.,*

```python
def sparsity_estimator(sigma):
    return (1/sigma) * np.sqrt(2 * np.log(sigma))
sigma_est= K*np.sqrt(np.mean((z-np.mean(z))**2))
kappa_bar_est = K * sparsity_estimator(sigma_est)
```

In Figure 3-right panel, we see that the actual $\bar{\kappa}$ and the plug-in approximation are quite close. Also see Figure 6 for usage of the formulation in a real world task.

**Remark 7.** *Although the Gaussian-type assumption is only for the sake of analysis and may not be valid in practice, we remark that the assumption is common in the machine learning and statistics literature. For example, high-dimensional random feature maps theoretically behave like Gaussian random variables at the thermodynamic limit (Goldt et al., 2022). Moreover (Tang et al., 2024, Figure 1) observed that logits in large language models can be modeled as Gaussian distributed plus outliers.*

## 4 Efficient implementation

The core challenge in computing the familywise loss lies in identifying in the forward pass the top-$\kappa$ logits that maximize the $\kappa$-th SHANCS $H_\kappa$. Performing a full sort to compute the loss becomes computationally expensive as the number of classes $K$ increases. To eliminate the need for sorting, we introduce a scalable retrieval-style algorithm based on a *heap-of-heaps* data structure.

In the backward pass, gradients are also computed efficiently within a memory initialization routine that leverages cached values from the forward pass. Together, these techniques make the familywise

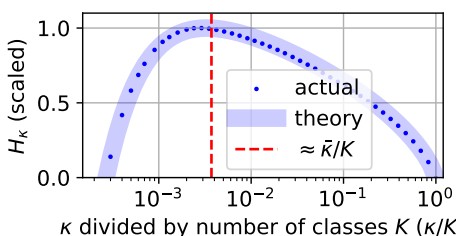 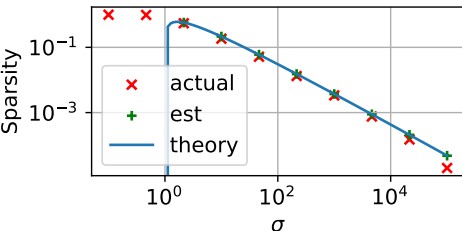

Figure 3: **(Left)** Approximate versus actual $\bar{\kappa}$ from simulated logits $\mathbf{z} \sim \mathcal{N}(0, 0.1)$. We set $K = 10000$ and (and thus $\sigma = 1000$ in the notation of Theorem 5). For each blue dot, the x-coordinate is $\kappa$ while the y-coordinate corresponds to $H_{\kappa}$. We subsample the scatter points to avoid visual clutter. The thick blue ribbon is the limiting curve as $K \to \infty$. **(Right)** $\bar{\kappa}$ versus $\sigma$. The blue line represents the theoretical prediction from Theorem 5, where the actual $\sigma$ is used.

loss practical for tasks with millions of classes, achieving wall-clock performance comparable to state-of-the-art cross-entropy implementations. See Figure 2. Below, we describe the implementation at a high level.

### 4.1 LOSS COMPUTATION VIA HEAP-OF-HEAPS SELECTION

Instead of sorting all $K$ logits in $\mathcal{O}(K \log K)$ time, we partition them into $C$ chunks, build a max-heap within each chunk, and insert the local max-heaps into a global max-heap to retrieve the next largest value across chunks. The high-level algorithmic idea for computing the loss per sample during the forward pass is as follows:

**Input:** Logits $z[1 \dots K]$, Target $y$
Partition logits into $C$ chunks;
**for** *each chunk* **do**
    Build a local max-heap;
    `// optionally store only top-M values if K is large`
**end**
**while** $H_{\kappa}$ *increases* **do**
    Pop max from global heap;
    Update `sum` of top-$\kappa$;
    Compute $H_k = (\texttt{sum} - 1)/k$;
    Track $\arg \max \bar{\kappa}$ and threshold $z_{\min} = z_{(\bar{\kappa})}$;
**end**
Compute loss: $1 - z_y + H_{\bar{\kappa}}$;

Each sample in a mini-batch is processed in parallel. The optimal $\bar{\kappa}$ and threshold $z_{\min}$ of each sample are cached for use in the backward pass.

**Adaptive heap truncation.** To further reduce preprocessing cost for extreme $K$, we limit the number of values retained in each local heap to a configurable threshold (starting at 100). If the retrieved values are insufficient to determine the optimal $\bar{\kappa}$, the threshold is increased (by a factor of 10) and the forward pass is restarted. Once the threshold reaches 10,000, the algorithm falls back to the exact routine with full heaps to ensure correctness.

**Complexity.** The algorithm processes each of the $K$ logits once to construct local heaps, requiring $\mathcal{O}(K)$ time. The top-$k$ search proceeds via repeated extraction from a global heap of size $C$. At each iteration, the global heap returns the next-largest logit across chunks in $\mathcal{O}(\log C)$ time, and the corresponding local heap is updated in $\mathcal{O}(\log(K/C))$ time. Together, this results in an effective per-iteration cost of $\mathcal{O}(\log K)$, and a total top-$k$ search cost of $\mathcal{O}(\bar{\kappa} \log K)$. In the worst case, $\bar{\kappa} = K$, giving a total runtime of $\mathcal{O}(K \log K)$. However, since $H_k$ is unimodal and typically maximized at

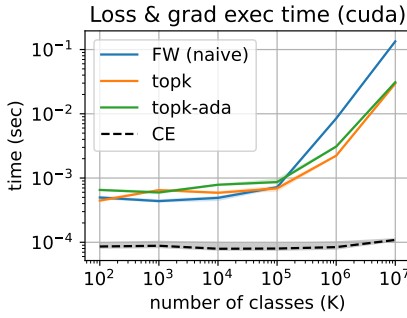 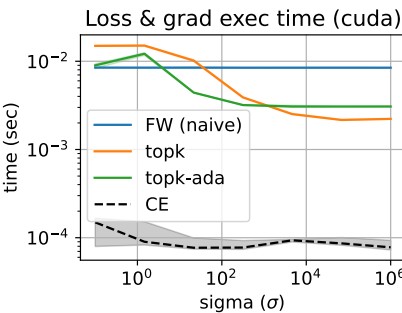

Figure 4: Similar to Figure 2 but on GPU. A loss-and-grad calculation speed comparison between the naive implementation of the FW loss (Appendix D), topk-based implementations (See Section 4.3) , and heaps based implementation, and the CE loss of PyTorch. Each run is repeated 20 times. The 20-80 quantiles are plotted as fill-betweens. **(Left)** Execution time versus the number of classes. **(Right)** Execution time versus $\sigma$ where $\sigma$ is proportional to the standard deviation of the logits as in Theorem 5.

$\bar{\kappa} \ll K$, the loop often terminates early. As a result, the observed wall-clock time grows more slowly than linear in $K$, a behavior we describe as *effectively sublinear*.

## 4.2 GRADIENT COMPUTATION VIA SINGLE-PASS INITIALIZATION

During backpropagation, the gradient of the familywise loss is computed efficiently over the mini-batch using a single-pass memory initialization procedure. For each sample in the mini-batch, all logits greater than or equal to the cached threshold $z_{(\bar{\kappa})}$ receive a uniform positive gradient of $1/(N\bar{\kappa})$, where $N$ is the mini-batch size. The target class logit is penalized by $-1/N$.

Given the cached values $(\bar{\kappa}, z_{\bar{\kappa}})$ from the forward pass, the gradient for logit $z_{ij}$ (where $i$ and $j$ are the sample and class index, respectively) in the mini-batch is defined as:

$$\frac{\partial \mathcal{L}}{\partial z_{ij}} = \begin{cases} 1/N\,\bar{\kappa}_i & \text{if } z_{ij} \geq z_{\min,i}, \\ -1/N & \text{if } j = y_i, \end{cases}$$

and $\frac{\partial \mathcal{L}}{\partial z_{ij}} = 0$ everywhere else. The gradient rule is applied independently to each sample in the mini-batch. The backward kernel processes all $N$ samples in a single streaming pass over their logits, yielding a total time complexity of $\mathcal{O}(NK)$ for the entire mini-batch.

| #datasets where ___ loss won | #features | |
|---|---|---|
| | $< 30$ | $\geq 30$ |
| FW | 22 | **15** |
| CE | **44** | 9 |
| Both | 20 | 10 |
| Total | 86 | 34 |

Table 1: Performance of familywise loss vs cross entropy loss-trained linear classifiers on the 90 datasets panel subset of the UCI dataset from Arora et al. (2020)

## 4.3 FASTER PURE PYTORCH IMPLEMENTATIONS

This concludes the description of the multicore CPU implementation of the FW loss. For efficient GPU implementations, we use the `torch.topk` operator as the basic building block. Moreover, we introduce an adaptive variant that uses Theorem 5 to guess the true $\bar{\kappa}$ as a "warm start". We term these two variants as "topk" and "topk-ada".

Our kernel provides significant speed up over the naive implementation. However, our fastest GPU kernels still do not beat cross entropy which has been highly optmized for the GPU. See Figure 4. However, we emphasize that training with FW loss led to more improvement per iteration. Thus, from a wall-clock versus performance point of view, FW loss should be considered superior to the cross entropy ( Figure 1 and Figure 5 ).

## 5 EXPERIMENTS

We demonstrate the efficacy of the familywise loss on the MNIST LeCun et al. (1998), CIFAR-100 Krizhevsky et al. (2009), and UCI datasets. In particular, we consider the 120 UCI datasets[2] is curated by Fernández-Delgado et al. (2014). We demonstrate that the FW loss outperforms cross entropy when we opt for parameter-free learning methods DoG (Ivgi et al., 2023), GDTUO (Chandra et al., 2022) and Prodigy (Mishchenko and Defazio, 2024). This section describes only the essential details of the experiments. For full details, see Appendix C.

### 5.1 UCI DATASETS

We use the Prodigy optimizer (Mishchenko and Defazio, 2024) with default settings to train linear model with the cross entropy and familywise loss without regularizer. Additional information about this experiment and the collection of datasets can be found in Appendix C.1.

For each loss choice and each dataset, we run five independent trials using random initializations. For each dataset, we use the original training/testing sets, and further split the training set 80/20 to obtain a validation set. For each run, we compute the test accuracy at the highest iteration with the highest validation accuracy. For stability, the final aggregate performance is taken to be the median across the five runs.

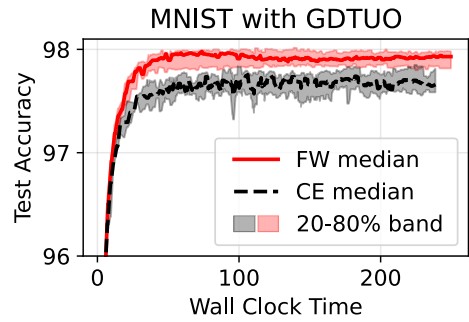

Figure 5: Test accuracy vs wall clock time on MNIST using the example code provided by GDTUO (Chandra et al., 2022).

For concreteness, we choose a threshold of #features $\geq 30$ to be considered a dataset with "large" feature space. Overall, familywise loss-trained linear classifiers appeared to perform better when the number of features is larger although the result is not statistical significant (CI = (0.464, 0.990) by Wald Test). On the other hand, cross entropy loss-trained linear classifiers did better on lower dimensional feature space datasets (CI = (0.540, 0.825) by Wald Test). See Table 1 for tabulated results of the benchmarking. Details on the benchmarking and statistical tests are in Appendix C.1.

### 5.2 MNIST USING "GRADIENT DESCENT: THE ULTIMATE OPTIMIZER" (GDTUO)

Chandra et al. (2022) introduces "Gradient Descent: The Ultimate Optimizer" (GDTUO) algorithm for learning learning rate schedules automatically by optimizing the learning rates on the fly using another iterative method. The official PyTorch implementation[3] contains an example training recipe of a two-layer fully-connected neural network for classifier MNIST digits. Apart from adding familywise loss an alternative to the cross entropy loss, we make no other adjustment to the recipe.

For each loss function, we perform 10 independent runs. In Figure 5, we plot the test accuracy curve versus the wall clock time over 5 epochs, as specified in the provided recipe. The familywise loss-trained model's final median accuracy is 97.93% while the cross-entropy-trained model's final median accuracy is 97.66%.

### 5.3 CIFAR-100 USING DoG

Next, we consider training a ResNet18 on the CIFAR-100 dataset. Unfortunately, we ran into numerical stability issues when using GDTUO, possibly due to the larger number of classes. Instead, we consider the DoG optimizer (Ivgi et al., 2023), which is also designed to selects learning rate and scheduler automatically, but using a different type of approach than GDTUO.

The DoG optimizer itself has a set of "hyper-hyperparameter". We follow the optimizer hyperparameter choices exactly as in the training recipe from the official GitHub repository of ' DoG[4]. For

---

[2]The original count is 121 datasets. However, the NURSERY dataset had corrupted labels, which we exclude.

[3]https://github.com/kach/gradient-descent-the-ultimate-optimizer/

[4]`example.py` from https://github.com/formll/dog/

non-optimizer related hyperparameters, since the published training recipe is for MNIST, we use commonly adopted settings for CIFAR-100 (e.g., batch size = 128, total number of epochs = 200).

We see that the model trained with FW loss starts off weak but eventually exceeds the model trained with the CE loss. Moreover from Figure 6, we see that as the test accuracy increases, the $\bar{\kappa}$ decreases. This is supports the intuition that $\bar{\kappa}$ can be thought of as confidence or a "neighborhood" around the true label.

## 6  DISCUSSION

This paper continues the study of familywise loss initiated by Duchi et al. (2018); Fathony et al. (2016) as an effective alternative to cross-entropy in large-scale classification. While earlier papers focused on statistical theory, our work focuses theory for efficient computation. Below, we discuss future directions.

Figure 6: Red curve denotes a run with the same set up as in Figure 1. Here we also plot the approximate and real $\bar{\kappa}$. The approximation uses the formula in Theorem 5.

**Automatic switching between sparse and dense computation**   There is a revival in interest in sparsity for deep learning (Martins and Astudillo, 2016; Nikdan et al., 2023). However, it is folklore that sparsity only pays off at extreme level of sparsity. Here, we observed during training that the familywise loss produces gradients that naturally become increasingly more sparse. This is in contrast to post-hoc gradient pruning methods that modifies a (dense) gradient into a sparse "pseudo"-gradient. Our theory result in Section 3.2 allows estimation of the sparsity of the gradient and opens the door to detect the sparsity threshold for switching to sparse matrix-vector multiply dynamically.

Sparsity may also be helpful for alleviating memory burden large language models pretraining when the number of classes, i.e., the vocabulary size, is large Wijmans et al. (2025). Moreover, there is increasing evidence that increasing the size of the vocabulary can be helpful. See Geiping and Goldstein (2023, Fig. 4) which demonstrates that the mean GLUE score increases with vocabulary size in a BERT-variant and more recently Tao et al. (2024) for larger language models. Future work will explore the role of familywise loss in the vocabulary-size scaling regime.

**Efficient GPU implementation.**   While this work focuses on high-throughput CPU execution, many real-world extreme classification scenarios such as recommendation systems and LLM pretraining are GPU-based (Rajput et al., 2023). The heap-of-heaps selection algorithm is well suited to CPUs due to its pointer-based structure and adaptive control flow. Implementing this logic efficiently on GPUs, however, requires data structures aligned with the GPU's massively parallel and SIMD-type architecture. In dense scenarios, array-based primitives are generally more natural on GPUs. For example, CUDA implementations could leverage device-level primitives such as `cub::DevicePartition` or segmented sorting. Our preliminary experiments indicate competitive runtime performance for class counts exceeding $10^5$. Nonetheless, full partitioning can be unnecessarily expensive in this context. familywise loss requires only the top-$\bar{\kappa}$ values, where $\bar{\kappa}$ is typically much smaller than $K$, but standard partitioning routines still move all elements. Developing custom GPU kernels that efficiently support this kind of truncated, retrieval-style selection remains an open and important direction for future work.

**Recovering class conditional distribution.**   This work evaluates the familywise loss primarily through the lens of classification accuracy. However, unlike cross-entropy, the familywise loss does not estimate the class conditional distribution, and as such, it may underperform when accurate class conditional distribution is needed. Metrics such as perplexity, which are widely used in language modeling and other generative settings, rely on well-calibrated probability estimates. In probability calibration literature, the link function Williamson et al. (2016) transforms the model logits into probabilities. For model trained by the cross entropy loss, the link function is the softmax. For the familywise loss, the procedure for transforming logits into probabilities is unknown. Future research on multiclass probability calibration, e.g., Berta et al. (2024), could potentially shed light on this.

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

# A   PROOF OF RESULTS FROM SECTION 3

## A.1   PROOF OF PROPOSITION 3

We restate the proposition for the reader's convenience.

**Proposition 8** (Bitonicity). *Let $z_1, \ldots, z_K \in \mathbb{R}$ be arbitrary numbers and $z_{(1)} \geq \cdots \geq z_{(K)}$ be these numbers sorted in descending order. Define $R_\ell := z_{(1)} + \cdots + z_{(\ell)} = \sum_{j=1}^{\ell} z_{(j)}$ and $H_\ell := \frac{1}{\ell}(R_\ell - 1)$.*

*Then $H_1, H_2, \ldots$ is* unimodal, *i.e., there exists $m^* \in \{1, \ldots, K\}$ such that*

$$H_1 \leq \cdots \leq H_{m^*-1} \leq H_{m^*} \geq H_{m^*+1} \geq \cdots \geq H_K.$$

*Proof.* To prove the proposition, we note that the result is equivalent to the following statement: There exists at most one $m \in \{1, \ldots, K-1\}$ such that

$$(H_{m+1} - H_m)(H_m - H_{m_1}) < 0$$

For simplicity, let's write $s_\ell := z_{(\ell)}$. Now, we calculate $H_{m+1} - H_m$:

$$H_{m+1} - H_m = \frac{R_{m+1} - 1}{m+1} - \frac{R_m - 1}{m} \tag{7}$$

$$= \frac{m(R_m + s_{m+1}) - m - (m+1)R_m + (m+1)}{m(m+1)} \tag{8}$$

$$= \frac{m s_{m+1} - R_m + 1}{m(m+1)}. \tag{9}$$

Below, we focus on the above numerator. The claim is that

$$\delta_m := m s_{m+1} - R_m + 1$$

is non-increasing as $m$ increases, and hence the sequences $\delta_1, \delta_2, \ldots$ changes sign only once. To see this, note that

$$
\begin{aligned}
\delta_{m+1} - \delta_m \\
&= (m+1)s_{m+2} - R_{m+1} + 1 - (m s_{m+1} - R_m + 1) \\
&= (m+1)s_{m+2} - s_{m+1} - m s_{m+1} \\
&= (m+1)s_{m+2} - (m+1)s_{m+1} \\
&= (m+1)(s_{m+2} - s_{m+1}) \leq 0
\end{aligned}
$$

Thus, $\delta_m$ is non-increasing in $m$. $\qquad\square$

## A.2   PROOF OF THEOREM 4

The technique we will use is essentially built around binary search with at most $T \leq \lceil \log_2(K) \rceil$ iterations, i,e, maximum depth. Let $a$ and $b$ be integers. Throughout this proof, we use the notation $[a, b]$ to denote the set $\{a, a+1, \ldots, b\}$. Similarly, we write $(a, b]$ to denote $\{a+1, \ldots, b\}$.

To avoid rounding-related issues, we assume that $K = 2^p$ is a power of 2. We will use $t$ as the iteration counter and we will see that the computational complexity of the $t$-th iteration is $O(K/2^t)$. Initially, we put $t = 0$. Thus, the total complexity is

$$O(2^p + 2^{p-1} + 2^{p-2} + \cdots + 1) = O(2^p) = O(K).$$

Below, we frequently mark a specific quantity with the superscript "$[t]$" to denote that the quantity was constructed during the $t$-th iteration. The algorithm will continuously modify the input sequence into the desired sequence without completely sorting. At the $t$-th iteration, sequence will be denoted by

$$\mathbf{z}^{[t]} := (z_1^{[t]}, z_2^{[t]}, \ldots, z_K^{[t]}) \quad \text{for } t = 0, 1, 2, \ldots, T.$$

Thus the original input sequence is denoted by $\mathbf{z}^{[0]}$. Let $L^{[t]}$ and $U^{[t]}$ be pointers that point to the beginning and end positions of the subvector operated the on during the binary search. Initially we

let $L^{[0]} = 1$ and $U^{[0]} = K/2$ be indices pointing to the "left half" of $z_1^{[0]}, \ldots, z_K^{[0]}$. For each iteration $t$, we define

$$\tilde{H}_\kappa^{[t]} = -\frac{1}{\kappa} + \frac{1}{\kappa} \sum_{j=1}^{\kappa} z_j^{[t]}$$

Note the difference between $\tilde{H}_\kappa^{[t]}$ and the earlier defined

$$H_\kappa = -\frac{1}{\kappa} + \frac{1}{\kappa} \sum_{j=1}^{\kappa} z_{(j)}.$$

Importantly, the cumulative sum in $\tilde{H}_\kappa^{[t]}$ is not assumed to be sorted. The sequence $\mathbf{z}^{[t]}$ will be defined such that the quantity $\tilde{H}_\kappa^{[t]}$ coincides with $H_\kappa$ at specific $\kappa$'s. The crucial ingredient is the `MidwayPartition` algorithm:

**Definition 9.** *The* `MidwayPartition` *algorithm takes as input an arbitrary set of numbers* $\mathbf{x} = (x_1, \ldots, x_N)$, *and a pair of indices $L$ and $U$ where $L < U$, and reorders the elements of $\mathbf{x}$ such that the reordered sequence $\mathbf{x}' = (x'_1, \ldots, x'_N)$ satisfies the following:*

*1. $x'_j = x_j$ for all $j \notin [L, U]$, i.e., entries outside the interval $[L, U]$ are unaffected,*

*2. $x'_{j_1} \geq x'_{j_2}$ for all $j_1 \in [L, (U+L)/2]$ and $j_2 \in ((U+L)/2, U]$.*

*More succinctly, we write* $\mathbf{x}' = \mathtt{MidwayPartition}(\mathbf{x}, L, U)$.

**Remark 10.** *It is possible to find the median of a length $N$ list in $O(N)$ time Blum et al. (1973). Using Hoare partition, we can divide a list of length $N$ for each $\kappa \in \{1, \ldots, K\}$. into a* upper *and* lower *sublists.*

Since $\kappa H_\kappa + 1 + z_{(\kappa+1)} = \sum_{j=1}^{\kappa+1} z_{(j)}$, we observe that

$$H_{\kappa+1} = -\frac{1}{\kappa+1} + \frac{1}{\kappa+1}(\kappa H_\kappa + 1 + z_{(\kappa+1)}) \tag{10}$$

**1st step**: Define $\mathbf{z}^{[1]} := (z_1^{[1]}, z_2^{[1]}, \ldots, z_K^{[1]}) = \mathtt{MidwayPartition}(\mathbf{z}^{[0]}, L^{[0]}, U^{[0]})$. We observe that

$$\tilde{H}_{K/2}^{[1]} = H_{K/2} \tag{11}$$

since the subsequence $z_1^{[1]}, \ldots, z_{K/2}^{[1]}$ is the same as $z_{(1)}, \ldots, z_{(K/2)}$ up to reordering. Moreover, we can calculate $z_{(K/2+1)}$ by taking the maximum over the sequence $z_{K/2+1}^{[1]}, \ldots, z_K^{[1]}$. Using Equation (10), we have successfully calculated $H_{K/2}$ and $H_{K/2+1}$ with only a partition and a max operation over a list of length $K$.

Moreover, we have enough information to determine whether $\bar{\kappa} \leq K/2$ or $\bar{\kappa} \geq K/2$. By Proposition 3, we deduce that if $H_{K/2} \geq H_{K/2+1}$, then $\bar{\kappa} \leq K/2$. Otherwise, $\bar{\kappa} \geq K/2$. Based on this, we can either "go left" or "go right" by updating our pointers $L^{[0]}$ and $U^{[0]}$ to be the boundary of either the left or right sublist, i.e.,

$$L^{[1]} := L^{[0]}, \ U^{[1]} := K/2 \quad , \text{or} \quad L^{[1]} := K/2, \ U^{[1]} := U^{[0]} \tag{12}$$

We can generalize this procedure to calculate values of $H_\kappa$ for other $\kappa$'s, i.e., the identity

$$\tilde{H}_\kappa^{[t]} = H_\kappa \tag{13}$$

for $t > 1$ for certain choices of $\kappa$'s. See the "induction step" part below for the formal statement, which requires a few definitions and lemmas to write down precisely. Toward this end, we start with a definition:

**Definition 11.** *The* step-$t$ condition *on the triplet* $(\mathbf{z}^{[t]}, L^{[t]}, U^{[t]})$ *holds if all of the following are true:*

*1. For every $j < L^{[t]}$ and every $j \geq L^{[t]}$, we have $z_j^{[t]} \geq z_{j'}^{[t]}$.*

2. *For every $j > U^{[t]}$ and every $j \leq U^{[t]}$, we have $z_j^{[t]} \leq z_{j'}^{[t]}$.*

3. $z_{j_1}^{[t]} \geq z_{j_2}^{[t]}$ *for all $j_1 \in [L^{[t]}, (U^{[t]} + L^{[t]})/2]$ and $j_2 \in ((U^{[t]} + L^{[t]})/2, U^{[t]}]$.*

By a similar argument as in Equation (11), if step-$t$ condition on the triplet $(\mathbf{z}^{[t]}, L^{[t]}, U^{[t]})$ holds, then

$$\tilde{H}_{(U^{[t]}+L^{[t]})/2}^{[t]} = H_{(U^{[t]}+L^{[t]})/2}. \tag{14}$$

The following lemma is a direct consequence of Definition 11 and the definition of `MidwayPartition`:

**Lemma 12.** *Suppose the step-$t$ condition (Definition 11) holds for $(\mathbf{z}^{[t]}, L^{[t]}, U^{[t]})$. Suppose one of the following two (mutually exclusive) conditions*

1. **Going left:** *Define $L^{[t+1]} := L^{[t]}$ and $U^{[t+1]} := (L^{[t]} + U^{[t]})/2$.*

2. **Going right:** *Define $L^{[t+1]} := (L^{[t]} + U^{[t]})/2 + 1$ and $U^{[t+1]} := U^{[t]}$.*

*Let $\mathbf{z}^{[t+1]} :=$ `MidwayPartition`$(\mathbf{z}^{[t]}, L^{[t+1]}, U^{[t+1]})$. Then the step-$(t+1)$ condition (Definition 11) holds for $(\mathbf{z}^{[t+1]}, L^{[t+1]}, U^{[t+1]})$.*

**Induction step**: Given a triplet $(\mathbf{z}^{[t]}, L^{[t]}, U^{[t]})$ that satisfies the step-$t$ condition (Definition 11). Suppose that Equation (14) holds. If we define $(\mathbf{z}^{[t+1]}, L^{[t+1]}, U^{[t+1]})$ using either the "going left" or the "going right" cases in Lemma 12, then both

$$\tilde{H}_{(U^{[t+1]}+L^{[t+1]})/2}^{[t+1]} \quad \text{and} \quad \tilde{H}_{(U^{[t+1]}+L^{[t+1]})/2+1}^{[t+1]} \tag{15}$$

can be computed in $O(U^{[t]} - L^{[t]} + 1)$ time.

To see why the induction step holds, let us first consider the "going left" case. For simplicity, let us write $\kappa_t := (U^{[t]} + L^{[t]})/2$. Note that

$$\kappa_t \tilde{H}_{\kappa_t}^{[t]} - \kappa_{t+1} \tilde{H}_{\kappa_{t+1}}^{[t+1]} = \sum_{j=\kappa_{t+1}+1}^{\kappa_t} z_j^{[t+1]}$$

Thus, to compute $\tilde{H}_{\kappa_{t+1}}^{[t+1]}$ given $\tilde{H}_{\kappa_t}^{[t]}$, we simply have to compute the sum $\sum_{j=\kappa_{t+1}+1}^{\kappa_t} z_j^{[t+1]}$ which consists of at most $\kappa_t$ terms. Next, to compute $\tilde{H}_{\kappa_{t+1}+1}^{[t+1]}$, we need to compute $\max_{j=\kappa_{t+1}+1}^{\kappa_t} \{z_j^{[t+1]}\}$ which can be done in $O(\kappa_t)$.

Now to complete the proof, we must make a decision at every iteration $t$ of either going left or going right. This is done in an analoguous fashion as in the paragraph immediately preceding Equation (12) in the base case. The procedure halts once $L^{[t]} = U^{[t]}$ which occurs in $t = \lceil \log_2 K \rceil$ steps. $\square$

# B   PROOF OF THEOREM 5

Let $\mathbf{z} \in \mathbb{R}^K$ be inputs to the familywise loss and suppose that entries $z_j$ of $\mathbf{z}$ are distributed according to normal distribution. In other words

$$z_j \overset{\text{iid}}{\sim} \mathcal{N}(0, \sigma_K), \quad j = 1, \dots, K$$

where $\sigma_K = \sigma_0/K$ and $\sigma_0 > 0$ is a constant. Recall that the "index-in-parenthesis" notation $z_{(j)}$ denotes the entries of $\mathbf{z}$ sorted in descending order, i.e.,

$$z_{(1)} \geq z_{(2)} \geq \cdots \geq z_{(K)}.$$

Let $\kappa \in \{1, \dots, K\}$ denote an arbitrary index. Our goal is to approximate the $\kappa$ that maximizes the $\kappa$-SHANCS as $K \to \infty$, i.e.,

$$\bar{\kappa} := \underset{\kappa=1,\dots,K}{\operatorname{argmax}} \; -\frac{1}{\kappa} + \frac{1}{\kappa} \sum_{j=1}^{\kappa} z_{(j)}.$$

To this end, we view the $\kappa$-th SHANCS as a function of $\kappa/K$. Intuitively, the function can be visualized by plotting $\frac{\kappa}{K}$ on the X-axis and the $\kappa$-th SHANCS $-\frac{1}{\kappa} + \frac{1}{\kappa} \sum_{j=1}^{\kappa} z_{(j)}$ on the Y-axis. In other words, we plot the following scatter points

$$\left( \frac{\kappa}{K}, \; -\frac{1}{\kappa} + \frac{1}{\kappa} \sum_{j=1}^{\kappa} z_{(j)} \right), \qquad \kappa = 1, \ldots, K.$$

See Figure 3-left panel blue scatter points for the resulting plot. As $K \to \infty$, this scatter plot converges to a function over the unit interval $[0, 1]$ shown by the light blue colored thick band in Figure 3-left panel. Below, we will

1. calculate this limiting function,
2. show that the limiting function attains exactly one argmax for any $\sigma_0$, and
3. derive an approximation of the argmax of the limiting function.

If $\kappa$ is constant as $K \to \infty$, then $\kappa/K \to 0$. Thus, we perform the calculation assuming that $\kappa/K \to p \in (0, 1)$ for some fixed value $p$ as $K \to \infty$.

### B.1 CALCULATING THE LIMITING FUNCTION

We begin with a simpler calculation of the limiting curve as $K \to \infty$ of

$$\left( \frac{\kappa}{K}, \; \frac{z_{(\kappa)}}{\sigma_K} \right), \qquad \kappa = 1, \ldots, K. \tag{16}$$

Let $\Phi$ be the cumulative distribution function (CDF) of the standard Gaussian/normal distribution (with zero mean and unit variance, denoted by $\mathcal{N}(0, 1)$). The *quantile function*, denoted $\Phi^{-1}$, is the (function-theoretic) inverse of $\Phi$.

By assumption, we have that the $z_j/\sigma_K$, $j = 1 \ldots, K$, are distributed according to $\mathcal{N}(0, 1)$. Moreover, $z_{(\kappa)}/\sigma_K$ converges to the $(1 - p)$-th quantile as $K \to \infty$. Therefore, we have that $\lim_{K \to \infty} z_{(\kappa)}/\sigma_K = \Phi^{-1}(1 - p)$ and the graph defined by Equation (16) converges as $K \to \infty$ to

$$(p, \Phi^{-1}(1 - p)).$$

Apply the "running average" operator to Equation (16), we have

$$\left( \frac{\kappa}{K}, \; \frac{1}{\kappa} \sum_{j=1}^{\kappa} \frac{z_{(j)}}{\sigma_K} \right), \qquad \kappa = 1, \ldots, K.$$

Multiplying and dividing by $K$, we have

$$\frac{1}{\kappa} \sum_{j=1}^{\kappa} \frac{z_{(j)}}{\sigma_K} = \frac{K}{\kappa} \sum_{j=1}^{\kappa} \frac{z_{(j)}}{\sigma_K} \cdot \frac{1}{K} \quad \to \quad \frac{1}{p} \int_0^p \Phi^{-1}(1 - t) dt$$

Since $\sigma_K := \sigma_0/K$ by definition, we have

$$K \left( \frac{1}{\kappa} \sum_{j=1}^{\kappa} z_{(j)} \right) \quad \to \quad \frac{\sigma_0}{p} \int_0^p \Phi^{-1}(1 - t) dt.$$

Moreover, using the fact that $\kappa/K \to p$, we have

$$K \left( -\frac{1}{\kappa} + \frac{1}{\kappa} \sum_{j=1}^{\kappa} z_{(j)} \right) \quad \to \quad -\frac{1}{p} + \frac{\sigma_0}{p} \int_0^p \Phi^{-1}(1 - t) dt.$$

The integral term on the right hand side is known as the expected shortfall of the normal/Gaussian distribution Khokhlov (2016) and can be expressed as

$$\frac{1}{p} \int_0^p F^{-1}(1 - t) dt = \frac{\varphi(\Phi^{-1}(p))}{p} =: \mathrm{ES}(p). \tag{17}$$

where $\varphi$ is the density function of the standard normal/Gaussian distribution. We have thus proven:

**Lemma 13.** *If the entries of $\mathbf{z} \in \mathbb{R}^K$ are distributed iid according to $\mathcal{N}(0, \sigma_K)$ where $\sigma_K := \sigma_0/K$. Suppose that $\kappa$ and $K$ both tend to $\infty$ such that $\kappa/K \to p \in (0,1)$ where $p$ is fixed. Then*

$$K\left(-\frac{1}{\kappa} + \frac{1}{\kappa}\sum_{j=1}^{\kappa} z_{(j)}\right) \quad \to \quad -\frac{1}{p} + \sigma_0 \text{ES}(p) \quad as \quad \kappa, K \to \infty.$$

### B.2 THE LIMITING FUNCTION HAS EXACTLY ONE ARGMAX

The goal of this subsection is to prove the following **claim**: For any $\sigma_0 > 0$, the function $g(p) := \sigma_0 \text{ES}(p) - (1/p)$ has exactly one maximizer on $(0, 1)$.

*Proof of claim.* We will show that $g'$, the first derivative of $g$, is initially strictly increasing, then becomes strictly decreasing. To simplify notation, let $c = \sigma_0$. For convenience, let $w = \Phi^{-1}(p)$. By definition, we have

$$g(p) = \frac{c\varphi(w) - 1}{p}$$

By the quotient rule, we have

$$g'(p) = \frac{p \cdot \frac{d}{dp}(c\varphi(w) - 1) - (c\varphi(w) - 1)}{p^2}$$

By the chain rule:

$$\frac{d}{dp}(\varphi(w)) = \varphi'(w) \cdot \frac{dw}{dp} = \varphi'(w)/\varphi(w) = -w \tag{18}$$

where for the last identiy, we used the fact that for the standard normal distribution PDF $\varphi$ we have $-\varphi'(w) = -w\varphi(w)$ for any $w \in \mathbb{R}$. Therefore

$$g'(p) = \frac{p \cdot (-cw) - (c\varphi(w) - 1)}{p^2} = \frac{-pc\Phi^{-1}(p) - c\varphi(\Phi^{-1}(p)) + 1}{p^2}. \tag{19}$$

Now, we claim that $h(p) := -pc\Phi^{-1}(p) - c\varphi(\Phi^{-1}(p)) + 1$, i.e., the numerator of $g'$, is

1. strictly decreasing on $(0, 1)$,

2. $\lim_{p \to 0} h(p) = 1$, and

3. $\lim_{p \to 1} h(p) = -\infty$.

This will prove the desired claim. Now, to see item 1, we show that $h'$ is strictly negative:

$$\frac{d}{dp}(p \cdot (-cw) - (c\varphi(w) - 1)) = -cw - pc\frac{dw}{dp} - c\frac{d}{dp}\varphi(w).$$

Using Equation (18), we get $\frac{d}{dp}\varphi(w) = -w$ and so the above first and last term cancels out. Thus,

$$\frac{d}{dp}(p \cdot (-cw) - (c\varphi(w) - 1)) = -pc\frac{dw}{dp} = -pc\frac{1}{\varphi(\Phi^{-1}(p))}$$

by the inverse function theorem and the fact that $\frac{dp}{dw} = \frac{d}{dw}\Phi(w) = \varphi(w)$. Since $\varphi(\cdot) > 0$ is always positive, we have that proved that $h'$ is strictly negative and thus $h$ is strictly decreasing on $(0, 1)$.

For item 3, clearly, as $p \to 1$, we have that $h(p) \to -\infty$. This is because $\lim_{p \to \{0,1\}} c\varphi(\Phi^{-1}(p)) = c\varphi(\pm\infty) = 0$. Moreover, $\lim_{p \to 1} -pc\Phi^{-1}(p) = -c\lim_{p \to 1}\Phi^{-1}(p) = -c(+\infty) = -\infty$.

It remains to prove that, $\lim_{p \to 0} h(p) = 1$. To see this, note that we already showed $\lim_{p \to 0} c\varphi(\Phi^{-1}(p)) = 0$. Below, we show that $\lim_{p \to 0} -p\Phi^{-1}(p) = 0$. It is well known that

$$\Phi^{-1}(p) \sim -\sqrt{-2\ln(p)} \tag{20}$$

for $p$ near 0. See paragraph immediately following (Mächler, 2022, Eqn. (18)). Thus, $\lim_{p \to 0} p\Phi^{-1}(p) = -\sqrt{-2 \lim_{p \to 0} p^2 \ln(p)}$. Now, by L'Hôpital's rule, we have

$$\lim_{p \to 0} p^2 \ln(p) = \lim_{p \to 0} \frac{\ln(p)}{1/p^2} = \lim_{p \to 0} \frac{1/p}{-2/p^3} = \lim_{p \to 0} -p^2/2 = 0$$

as desired. □

### B.3 AN APPROXIMATION OF THE ARGMAX

**Lemma 14.** *The derivative of* ES*, the expected shortfall of the standard normal/Gaussian distribution as defined in Equation* (17)*, is given by*

$$\frac{d\text{ES}}{dp}(p) = \frac{\Phi^{-1}(1-p)p - \varphi(\Phi^{-1}(1-p))}{p^2}.$$

*Proof.* By a similar calculation as before in deriving Equation (19), we have that

$$\frac{d\text{ES}}{dp}(p) = \frac{-p\Phi^{-1}(p) - \varphi(\Phi^{-1}(p))}{p^2}.$$

To conclude, we note that $\Phi^{-1}(\bullet) = -\Phi^{-1}(1 - \bullet)$ and moreover that $\varphi(-\bullet) = \varphi(\bullet)$ □

Given the earlier **claim** "For any $\sigma_0 > 0$, the function $g(p) := \sigma_0 \text{ES}(p) - (1/p)$ has exactly one maximizer on $(0, 1)$" and the formula for the derivative in Lemma 14, we can now compute

$$\underset{p \in (0,1)}{\text{argmax}} \ \sigma_0 \text{ES}(p) - \frac{1}{p}.$$

To this end, we take derivative w.r.t $p$ of the above and solve for roots (which we know there is exactly one by the claim):

$$\sigma_0 \frac{\Phi^{-1}(1-p)p - \varphi(\Phi^{-1}(1-p))}{p^2} + \frac{1}{p^2} = 0$$

Simplifying, we have

$$\Phi^{-1}(1-p)p - \varphi(\Phi^{-1}(1-p)) = -\frac{1}{\sigma_0}. \tag{21}$$

The goal is to solve for $p$ in terms of $\sigma_0$. However, the normal CDF $\Phi$ and its inverse $\Phi^{-1}$ do not have closed-forms. Thus, we resort to approximations. Let $t := \Phi^{-1}(1 - p)$. Thus, $p = 1 - \Phi(t)$. Then

$$\Phi^{-1}(1-p)p - \varphi(\Phi^{-1}(1-p)) = tp - \varphi(t). \tag{22}$$

Our goal now is to replace the LHS of Equation (22) by a simpler expression purely defined in terms of $p$ which holds when $p$ is close to 0. We recall (Small, 2010, Eqn. (2.38)) which says the following

$$\frac{\varphi(t)}{t} \left( 1 - \frac{1}{t^2} + \frac{3}{t^4} - \frac{3 \cdot 5}{t^6} + \frac{3 \cdot 5 \cdot 7}{t^8} - \cdots \right) \sim 1 - \Phi(t) \quad \text{as} \quad t \to +\infty.$$

We will only be using the first two terms of the series approximation:

$$\frac{\varphi(t)}{t} \left( 1 - \frac{1}{t^2} \right) \approx 1 - \Phi(t) \quad \text{as} \quad t \to +\infty. \tag{23}$$

Recall earlier that we let $t := \Phi^{-1}(1 - p)$ and thus $p = 1 - \Phi(t)$. From Equation (23) we get

$$\frac{\varphi(t)}{t} \left( 1 - \frac{1}{t^2} \right) \approx p \quad \text{as} \quad t \to +\infty. \tag{24}$$

Isolating $\varphi(t)$, we get

$$\varphi(t) \approx pt \left( \frac{1}{1 - \frac{1}{t^2}} \right) \quad \text{as} \quad t \to +\infty. \tag{25}$$

Using the geometric series and truncating to the first two terms, the above RHS can be approximated by

$$pt \left( \frac{1}{1 - \frac{1}{t^2}} \right) = pt(1 + t^{-2} + t^{-4} + \cdots) \approx pt(1 + t^{-2})$$

from which we deduce that

$$\varphi(t) \approx pt + \frac{p}{t}$$

Now, Equation (22) simplifies as

$$\Phi^{-1}(1 - p)p - \varphi(\Phi^{-1}(1 - p)) = tp - \varphi(t) = -p/t$$

when $p$ is close to $0$. To finally eliminate $t$, we recall Equation (20) from earlier where $\Phi^{-1}(p) \sim -\sqrt{-2\log(p)}$ Since $t = \Phi^{-1}(1 - p) = -\Phi^{-1}(p)$, we finally get

$$\Phi^{-1}(1 - p)p - \varphi(\Phi^{-1}(1 - p)) = tp - \varphi(t) = p/\sqrt{-2\ln(p)}.$$

Note that throughout, we are interested in the approximation when $p$ is near $0$. This precisely corresponds to the case when $t \to +\infty$ in which case all approximations used above hold. Thus, we have proven:

**Lemma 15.** *When $p$ is near $0$, we have*

$$\Phi^{-1}(1 - p)p - \varphi(\Phi^{-1}(1 - p)) \approx \frac{p}{\sqrt{-2\ln(p)}}.$$

Recall that our original motivation is to solve for $p$ in terms of $\sigma_0$ in Equation (21). Given the above lemma, we only need to solve for $p$ in terms of $\sigma_0$ in the following equation:

$$\frac{p^2}{2\ln(1/p)} = \frac{1}{\sigma_0^2}. \tag{26}$$

**Lemma 16.** *An approximation to* (26) *is given by $p \approx \frac{1}{\sigma_0}\sqrt{2\ln(\sigma_0)}$.*

Note that the above Lemma together with immediately implies our desired result Theorem 5.

*Proof.* For notational convenience, we write $a = \frac{1}{\sigma_0}$. So the goal is to solve for $p$ in

$$\frac{p^2}{2\ln(1/p)} = a^2.$$

Rearranging, we see that the above is equivalent to

$$\ln(p) = -\frac{p^2}{2a^2}.$$

Letting $y := -\ln(p)$, we have

$$ye^{2y} = \frac{1}{2a^2}.$$

Further substituting $u = 2y$, we get

$$ue^u = \frac{1}{a^2}.$$

To solve the above, recall that the Lambert W function (Corless et al., 1997, Eqn. (1)) satisfies the equation

$$W(z)e^{W(z)} = z$$

So if we substitute $W(z) = u$, then we have

$$W(z)e^{W(z)} = z = W^{-1}(u) = \frac{1}{a^2}.$$

In other words, $u = W(\frac{1}{a^2})$. Substituting back $p$, we see that

$$p = \exp\left(-\frac{1}{2}W\left(\frac{1}{a^2}\right)\right) \tag{27}$$

It is well known (e.g., see (Corless et al., 1997, § 4.2)) that

$$W(z) = \ln(z) - \ln(\ln(z)) + o(1).$$

Plugging the above into (27), we have

$$p = \exp\left[-\tfrac{1}{2}\left(\ln\left(\tfrac{1}{a^2}\right) - \ln\left(\ln\left(\tfrac{1}{a^2}\right)\right)\right)\right] = \exp\left[\ln(a) + \ln\left(\sqrt{\ln\left(\tfrac{1}{a^2}\right)}\right)\right] \qquad (28)$$

$$= a\sqrt{\ln(1/a^2)} = a\sqrt{2\ln(1/a)} \qquad (29)$$

Substituting in $\sigma_0 = 1/a$, we get the desired approximation. $\qquad\square$

## B.4   PUTTING IT ALL TOGETHER

In view of Lemma 13 and Appendix B.2's main claim, we only have to solve for the (unique) root of

$$-\frac{1}{p} + \sigma_0 \mathrm{ES}(p).$$

By Lemma 14, this reduces to solving the Equation (21). Using the approximations in Lemma 15 and Lemma 16, we get the desired result.

## C   ADDITIONAL EXPERIMENTAL DETAILS

At the root of the [supplemental material folder], run the following

```
conda create -n gradientasretrieval python=3.9 -y
conda activate gradientasretrieval
pip install torch==2.4.1
cd trackexp; pip install .; cd ..;
cd pyfamilywise; pip install . cd ..
```

The familywise losses implementations used throughout this paper are included in `pyfamilywise` as a python package. The package `trackexp` is a lightweight experiment tracking tool.

### C.1   LINEAR CLASSIFIERS ON UCI DATASETS

**Detailed metadata and numerical results**. The full table of metadata for the 121 UCI datasets can be found at [`UCI-metadata.csv` in the supplemental materials.]

The full table of test accuracy comparison can be found at [`UCI-accuracy-table.csv` in the supplemental materials.]

**Statistical tests**. The Wald test is implemented according to the procedure described in

`https://stats.stackexchange.com/a/530465`

for rejecting the hypothesis that "FW-loss training does not beat CE-loss training". Conceptually, we can view this hypothesis as testing if we should reject that "a coin is not biased toward head". We include the code snippet below.

```python
import numpy as np
from scipy import stats

def Wald_Test(heads, tails):
    # this test follows the procedure described in
    # https://stats.stackexchange.com/a/530465
    x = heads
    n = heads + tails # number of total flips
    p_hat = x / n
    print(f"Point estimate p_hat = {x}/{n} = {p_hat:.3f}")
    z_critical = 1.96  # for 95% CI
    margin_error = z_critical * np.sqrt(p_hat * (1 - p_hat) / n)
```

```
13    ci_lower = p_hat - margin_error
14    ci_upper = p_hat + margin_error
15    print(f"95% CI: ({ci_lower:.3f}, {ci_upper:.3f})")
16    p_value = stats.binomtest(x, n, 0.5).pvalue
17    print(f"P-value for fairness test: {p_value:.3f}")
18    alpha = 0.05
19    if p_value < alpha:
20        print(f"Reject H0: Coin is significantly unfair (p < {alpha})")
21    else:
22        print(f"Fail to reject H0: Cannot conclude coin is unfair (p >= {
      alpha})")
```

**Hardware**. The experiments were run on a machine with an x86 architecture Intel Xeon w5-3435X CPU with 32 logical cores. The full batch of experiments (including 5 replicates) completed in under 12 hours. GPU is not used for this experiment.

**Reproducing the experiments**. Follow the instruction at the beginning of Appendix C, then follow the instruction in `UCI-experiments/README.md` in [the supplemental materials.]

## C.2  MNIST USING "GRADIENT DESCENT: THE ULTIMATE OPTIMIZER" (GDTUO)

**Reproducing the experiments**. Follow the instruction at the beginning of Appendix C, then follow the instruction in `MNIST/README.md` in [the supplemental materials.]

**Hardware**. The experiments were conducted on a NVIDIA RTX 5000 Ada on the same machine as in Appendix C.1. The full batch of experiments (including 10 replicates for each loss function) completed in under 3 hour.

## C.3  CIFAR-100

**Additional replicates**. We plot the results from four additional replicates for both the FW and CE losses. We use GPUs from different machines to compute these replicates so the wallclock time are not directly comparable. Thus Figure 7 plots the epochs on the X-axis.

**Hardware**. The experiments in the main article were conducted on a NVIDIA RTX 5000 Ada on the same machine as in Appendix C.1. The additional replicate experiments were run on two cloud computing instances with Quadro RTX 6000 GPUs. The full batch of experiments (including four replicates for each loss function) completed in under 12 hour.

**Reproducing the experiments**. Follow the instruction at the beginning of Appendix C, then follow the instruction in `CIFAR-100/README.md` in [the supplemental materials.]

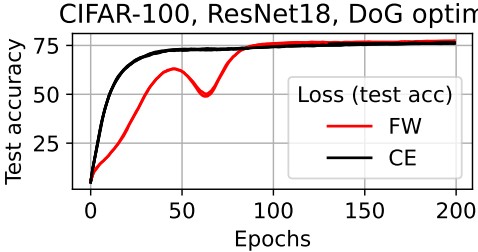

Figure 7: Test accuracy versus epoch on training ResNet18 using the DoG optimizer (Ivgi et al., 2023).

## D  NAIVE IMPLEMENTATION OF THE FAMILYWISE LOSS

```
z = logits # has shape (batch_size, num_classes)
y = target # has shape (batch_size)
z_at_y = z[torch.arange(z.shape[0]), y]
z_hnscs = torch.sort(z, axis=-1, descending=True).values
z_hnscs.cumsum_(axis=-1)
z_hnscs.sub_(1)
kappa_inv = 1/torch.arange(1,num_classes+1).to(device)
z_hnscs.mul_(kappa_inv[torch.newaxis,:])
losses = 1 - z_at_y + torch.max(z_hnscs, axis=-1).values
```

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
