# OpenReview forum: "Gradient-as-retrieval: Classification beyond Cross Entropy"
_ICLR.cc/2026/Conference — Submitted to ICLR 2026_

### Official Review · Reviewer_vUcr · 2025-10-23

**Soundness:** 3
**Presentation:** 3
**Contribution:** 3
**Rating:** 6
**Confidence:** 2

**Summary:**

The paper proposes Familywise (FW) loss as a practical alternative to cross-entropy (CE) for multiclass classification, framing its computation as “gradient-as-retrieval.” The key observation is that a subgradient depends only on the true class and the top-$\bar{\kappa}$ impostor logits, yielding the simple form

$$ \nabla_z L(z,y)= -\mathrm{one\text{-}hot}_y + \tfrac{1}{\bar{\kappa}}\ \mathrm{many\text{-}hot}_{{(1),\ldots,(\bar{\kappa})}}, $$

so the dominant operation is top-(k) selection rather than softmax/exponentials. The paper proves a bitonicity property for a sequence used to pick $\bar{\kappa}$ (enabling an $\mathcal{O}(K)$ subgradient algorithm) and presents a Gaussian-logits approximation to estimate $\bar{\kappa}$ and explain emergent gradient sparsity during training. Implementations (multicore CPU, pure PyTorch GPU) and experiments on UCI, MNIST, and CIFAR-100 with parameter-free optimizers (Prodigy, GDTUO, DoG) indicate that FW is competitive and often favorable in those regimes (e.g., MNIST median 97.93\% vs 97.66\% for CE). The paper also clarifies that FW avoids transcendental functions and does not directly provide calibrated probabilities like CE/softmax.

**Strengths:**

- **Practical relevance.** Computation hinges on selection/top-(k), avoiding transcendental function, attractive for efficiency- or encryption-constrained settings.

- **Sparsity perspective.** The Gaussian-logits analysis gives an interpretable link between logit variance $\bar{\kappa}$, and gradient sparsity, with qualitative support across class counts.

- **Working implementations.** CPU and PyTorch GPU paths exploit the theory; the CPU shows speedups vs CE at large ($K$) (with GPU improving over a naïve FW baseline).

- **Empirical validation in parameter-free regimes.** UCI/MNIST/CIFAR-100 results with Prodigy/GDTUO/DoG support the claim that FW is particularly competitive when schedules are learned automatically; MNIST numbers are concretely reported.

**Weaknesses:**

-  **Baseline strength and fairness.** The empirical focus is on parameter-free optimizers; this is a legitimate and interesting regime, but it leaves uncertainty about tuned CE (e.g., cosine decay, label smoothing, warmup). Without strong, standard CE baselines on MNIST/CIFAR-100, gains may conflate optimizer–loss interactions with the inherent merits of FW.

- **Statistical rigor and reporting.** Some results note non-significance for trends, but a consistent reporting of mean $\pm$ std across seeds, paired significance tests, and effect sizes is not visible throughout. CIFAR-100 is described largely qualitatively.

- **Approximation vs exact computation.** The Gaussian estimator for $\bar{\kappa}$ is elegant, but the empirical accuracy/runtime trade-off compared to exact selection is not quantified across ($K$) (notably at moderate ($K=100$) vs very large ($K$)).

- **Calibration and applicability limits.** The paper highlights the absence of calibrated probabilities under FW; however, there is no experiment illustrating calibration error, uncertainty estimation, or downstream impact where probabilities matter (e.g., selective classification).

- **Breadth of evaluation.** There is no study of *class imbalance/long-tail* or larger-scale datasets; these are common stress tests for alternative losses and would increase external validity.

**Questions:**

1. **Tuned CE baselines:** Can results on MNIST and CIFAR-100 include standard, tuned CE (e.g., cosine LR with warmup, label smoothing) to disentangle optimizer–loss coupling from the intrinsic effect of FW?

2. **Approximation ablation:** What is the accuracy vs runtime delta between exact $\bar{\kappa}$ selection and the Gaussian approximation for ($K=100$) and ($K\ge 10^4$)? Plotting both within the same wall-clock budget would clarify practical trade-offs.

3. **Sparsity dynamics:** Can the paper report $\bar{\kappa}/K$ over training (with accuracy) to substantiate the claimed increasing gradient sparsity and its correlation with performance?

4. **Calibration assessment:** How does FW compare to CE in calibration error (ECE/mECE), and is there a practical post-hoc mapping to obtain usable probabilities when needed?

6. **Robustness to imbalance/long-tail:** How does FW behave on long-tailed datasets or with focal-style reweighting compared to CE variants?

---

> ### Author Response · Authors · 2025-11-19
>
> We thank the reviewer for the constructive feedback.
>
> Regarding "Baseline strength and fairness", we believe that our methodology is more fair, giving a blank slate to both the CE and FW losses. If we used a method tailored to the CE loss, then it would not be surprising that CE does better. Our experiments level the playing field to compare the losses without using prior hand-tuning.
>
> Regarding "Statistical rigor and reporting", our CIFAR100 experiments are replicated 5 times (Figure 7 has 4 replicates + Figure 1 replicate). Our experiments on MNIST are also replicated 10 times.
>
> Regarding "Approximation vs exact computation", we test our sparsity estimator on large K, in figure 3, K = 10000 for synthetically logits. We plan to, in future work on applying FW loss to large-scale classification, estimate the approximation on real world experiments.
>
> Regarding "Calibration and applicability limits", this is a common mixup of "probability-calibration" and "classification-calibration". FW loss is specificially designed for "classification-calibration" (and importantly, not for probability-calibration). This is in fact the same distinction between hinge loss and binary cross entropy loss. A different type of technique is used to convert hinge loss-trained classifier to probabilities (e.g., Platt scaling). Examining this phenomenon for the FW loss is an interesting future direction which we will include a discussion of.
>
> Regarding "Breadth of evaluation" and long-tailed distribution, this is an interesting suggestion. We believe the question of modifying classification loss functions (including both the cross entropy and FW losses) are research questions of independent interest. We would expect techniques for addressing challenges from long-tailed tasks to equally apply to both cross entropy and FW loss. We will add this as a future direction.

---

### Official Review · Reviewer_oiVK · 2025-10-31

**Soundness:** 4
**Presentation:** 3
**Contribution:** 2
**Rating:** 6
**Confidence:** 2

**Summary:**

This submission studies the properties of familywise (FW) loss as an alternative to cross-entropy loss learning classification tasks. FW loss is free of transcendental functions (log, exp, etc.) and mainly relies on the computation of the top-k largest (pre-softmax) logits for some k that maximizes a simple function. The authors prove that this k (and therefore the FW loss and its gradients) can be computed in O(K) time (where K is the total number of logits) using a unimodality property and “retrieval-style” algorithms, hence giving a better theoretical bound than the naive O(K logK) sorting. They also show k can be approximated with easy-to-compute quantities when the logits follow a Gaussian distribution. Lastly, they provide an efficent-in-practice implementation of computing FW loss (and its gradient) using the heap-of-heaps data structure. They experimentally demonstrate the FW can outperform cross-entropy (in terms of time/accuracy) under certain settings, in particular when using CPUs / the number of features is large, but is still defeated by cross-entropy when using GPUs / the number of features is small.

**Strengths:**

1) The paper combines sound theoretical claims with extensive experiments. While proofs of theoretical claims are deferred to the appendix, proof sketches are provided, which is very useful for building intuition.

2) The authors do not shy away from discussing the limitations of their approach, such as underperforming with GPUs (and other methods that are largely optimized by cross-entropy loss) or the fact that FW does not estimate class conditional distributions. The discussion of such limitations is appreciated and makes it easier to comprehend the contributions of the paper.

3) The paper is overall well-written and easy to read. The figures display clear trends and shine light on the performance of FW vs cross-entropy loss.

**Weaknesses:**

1) As the authors admit, FW losses currently lack an efficient GPU implementation (the implementation introduced in this submission is optimized for CPUs) which inevitable limits the applicability of the paper considering how widely GPUs are adopted (e.g. see Fig. 4). While this indeed results from an unfair advantage to cross-entropy loss due wide adaptation (and therefore optimizers/architectures being tuned for it), an alternative loss (or another implementation of FW) that outperforms cross-entropy across different optimizers/architectures would be a much more attractive contender. While there is some discussion on the challenges of GPU implementation, it is not clear how much the results in this paper can be built on for that setting.

2) While the mathematical properties of FW loss is rigorously discussed, those of cross-entropy are largely left out of the paper. In fact, cross entropy itself or the classification-calibration property satisfied by both losses is never formally defined in the paper. This makes it hard to appreciate how much stronger the universal equivalence to 01 loss satisfied only by FW loss actually is. In general, since so much of the paper's motivation centers around finding an alternative to cross-entropy loss, a more extensive discussion of the mathematical properties of it (along with which of them are satisfied by FW loss also) would be beneficial.

3) Some statements/claims in the paper are anecdotal/unclear. For instance, Theorem 5 says "as $K\rightarrow \infty", $\overline{\kappa}$ can be approximated as XXX", but it is not clear what "can be approximated means" without any approximation bounds. Similarly, the claim "Since H_k is (...) typically maximized at $\overline{\kappa} << K$, the loop often terminates early" is not justified by any results/citations.

Minor:
- l30 "statistics, appeared" -> "statistics and appears"
- l35 "FW loss" -> familywise loss has not been mentioned in the introduction yet, so do not use an acronym.
- l75 "Theorem 5" should be in parentheses.
- l91 "While less widely known" -> than what?
- l91 "in (Bartlett" -> use citet rather than citetp
- l100 extra ")" inside both $\mathbb{E}_{\mathbf{X},Y}$. Also, the second $\mathbb{E}_{\mathbf{X},Y}$ should have $f$ rather than $f_n$.
- In general, the pages in which figures/tables are mentioned for the first time do not match the page they appear (and in fact do not follow the same order), which decreases readibility.
- l235 environment should be labeled "Proof sketch" rather than "Proof"
- l293 "Instead of sorting all $K$ logits in $O(K \log K)$ time..." -> this sentence seems to ignore/omit the results from Section 3. While you have talked about efficiency in practice there, it would still be worth mentioning here for the coherence of the paper.
- l375 "optmized" -> "optimized"
- l427 "set of 'hyper-hyperparameter' " -> "set of 'hyper-hyperparameters' "
- l438 "This is supports" -> "This supports"
- l473 "(...) context. familiywise loss requires only (...)" -> capitilize f
- l483 "Williamson et al (...)" -> use citep rather than citet.

**Questions:**

Can you discuss other desirable properties of cross-entropy that are still satisfied by FW loss, beyond classification-calibration?

---

> ### Author Response · Authors · 2025-11-19
>
> We thank the reviewer for the constructive feedback and the minor fixes.
>
> For weakness 1, we found that despite the FW loss being *per-iteration* slightly slower on the GPU, this did not affect the *overall* performance, even on the GPU. All of our experiments are ran on the GPU, where FW loss combined with hyper-hyperoptimizer, shows better performance than the CE loss. We hope our work will motivate the GPU programming community to design more efficient algorithms for the GPU such as SegmentedMaxHeap.
>
> For weakness 2, the statistical learning-theoretic advantages of FW loss is entirely developed in the prior work [1]. Our work focuses on the theory for supporting the efficient computation of the FW loss gradient, which is the novelty of our work. However, we do agree it would be beneficial to include such a discussion. We will modify our manuscript to reflect this feedback.
>
> For weakness 3, the approximation results are derived by truncating various Taylor series through the proof. Regarding the claim "Since H_k is (...) maximized for $\bar{k} \ll K$", Theorem 5 gives a scenario where observation holds. It is not always the case, that  $\bar{k} \ll K$, but we find this to be the case often in practice (Figure 6, black and gray curves).
>
> ---
>
> [1] Duchi, John, Khashayar Khosravi, and Feng Ruan. "Multiclass classification, information, divergence and surrogate risk." (2018): 3246-3275.

---

### Official Review · Reviewer_GTVe · 2025-11-01

**Soundness:** 2
**Presentation:** 2
**Contribution:** 3
**Rating:** 4
**Confidence:** 3

**Summary:**

This work is motivated by the expense of evaluating cross-entropy in fully homomorphic encryption due to transcendental functions, such as exponentials and logarithms. The familywise (FW) loss avoids transcendental operations, and this work develops new theoretical insights that enable efficient computation of FW gradients through retrieval-style algorithms. Because most existing optimizers and learning-rate schedules are tuned for CE, this work shows that FW surpasses CE in performance when used with parameter-free learning algorithms.

**Strengths:**

1. **Theoretical grounding**: Extends known properties of FW loss (Duchi et al., 2018) and connects them with efficient computational methods. The bitonicity property and $O(K)$ computation result are elegant and novel from an algorithmic standpoint.
2. **Computational innovation**: The heap-of-heaps and top-k retrieval implementations are nontrivial engineering contributions that make the loss practically usable for large-scale problems.

**Weaknesses:**

1. **Characterize the failure mode**: In Section 5.3, it says that using FW loss on CIFAR-100 has numerical stability issues when using gradient descent, possibly due to the larger number of classes. I think this failure mode is worth investigation: Can this hypothesis be validated on synthetic data problems, showcasing the relations between numerical stability or performance with the number of classes? Also, does it indicate its inapplicability to language tasks due to the large vocabulary size? More thorough discussions on this would be helpful.
2. **More non-linear architectures**: The introduction section claims that this work helps answer whether FW loss works in *deep learning* classifiers. Therefore, it would be helpful to provide experimental results with some small non-linear models in Sections 5.1 and 5.2. The models do not have to be large since the datasets are relatively small-scale, but it would be more complete if multiple widely used architectures are covered, including convolutional and transformer models. It would also be nice to cover architectures of different levels of complexity (in terms of depth). If FW loss works on different architectures of different complexities, the experiments would be convincing.
3. **Efficiency in privacy-preserving applications**: The paper claims that FW can be beneficial in fully homomorphic encryption for privacy-preserving applications where cross-entropy (CE) is expensive. To support this, it helps to present experimental results in privacy-preserving applications that compare the efficiency of FW to CE.

**Questions:**

There are some minor editing suggestions:
1. The word "transcendental" is used from the beginning without a brief introduction or synonyms as an explanation. It would be nice to add one when it's used for the first time.
2. It is mentioned that widely adopted optimizers and learning rate schedules are tuned to CE, which motivates the authors to use parameter-free learning methods. However, it still helps to report the FW results under widely adopted optimizers with their hyperparameters tuned for FW. If it underperforms, readers would be interested in at least seeing some discussions about this and having it left for future work.

---

> ### Author Response · Authors · 2025-11-19
>
> We thank the reviewer for the constructive feedback.
>
> Regarding the point about the "failure mode", we'd like to clarify that the issue existed for both the cross entropy and the familywise loss. Moreover, we'd like to clarify that our Sec 5.3 uses not the vanilla gradient descent, but rather GDTUO (gradient descent the ultimate optimizer) [1], which is a hyper-hyperoptimizer that automatically learns the learning rate. The intention behind this choice is that we wanted to compare the cross entropy with the familywise loss, using off-the-shelf learning rate tuner. The code was provided by [1]. We were unable to make GDTUO on CIFAR100 after changing the parameters of GDTUO. We believe this reflects a possible bug in GDTUO, since this failure mode occured for both cross entropy and familywise losses.
>
> More non-linear architecture: in our experiments, we do test on convolutional neural networks of various depth. We will update our manuscript with transformer-based architecture.
>
> Regarding FHE: In current FHE schemes (e.g., CKKS), the FW loss would indeed be quite expensive to implement due to the comparisons. However, existing FHE schemes are designed to support implementing low-degree polynomials. It is possible that a (possibly new) FHE scheme that go a different route would be better than losses like FW loss.
>
> Our contributions in this work are 1. theory that supports efficient computation of the gradient of FW loss and 2. demonstrate that hyper-hyperoptimizers can give advantage to the FW loss. Hence, regardless of whether such (possibly new) FHE scheme exist, we believe our work has merit. It is difficult to establish advantage of a new loss over the incumbent cross entropy from every aspects. In our work, we focused on the two aspects aforementioned, and hope our work to inspire new FHE schemes as a potential direction of future work.
>
> ---
>
> [1] Chandra, Kartik, et al. "Gradient descent: The ultimate optimizer." Advances in Neural Information Processing Systems 35 (2022): 8214-8225.

---

> > ### Comment · Reviewer_GTVe · 2025-11-26
> > **Response to Rebuttal**
> >
> > Thank you for the clarification on this paper's contributions and why GDTUO is not used on CIFAR-100. Except for this, my other concerns remain:
> > 1. This work is motivated by the benefit of FW in fully homomorphic encryption for privacy-preserving applications, but no experiment for privacy-preserving applications is presented.
> > 2. The experiments are too small-scale (the largest setting is ResNet18 on CIFAR-100).
> > 3. Currently, only results with optimizers GDTUO and DOG are presented. I am not convinced why the results of FW loss optimized by more commonly used optimizers are not presented. Hyperparameters can be tuned for each dataset/loss separately, and as a reader, I am curious about the results.
> >
> > Therefore, I would like to keep my score.

---

### Official Review · Reviewer_VjWb · 2025-11-02

**Soundness:** 2
**Presentation:** 3
**Contribution:** 1
**Rating:** 2
**Confidence:** 4

**Summary:**

This manuscript explores the use of the familywise (FW) loss function as an alternative to cross-entropy. Previous studies have discussed FW loss, but they have not examined its performance in deep learning classifiers nor proposed efficient computational methods for it. In this work, the authors present an approach to efficiently compute the gradients of the FW loss function and demonstrate that the performance of deep learning models using FW loss is comparable to that achieved with cross-entropy loss.

**Strengths:**

The part of the study that examines how performance changes when replacing the conventional cross-entropy loss is indeed an aspect that could attract the interest of AI researchers. Since cross-entropy loss has become almost a de facto standard loss function for deep learning classifiers, any attempt to replace it would be considered a challenging and ambitious topic in the research community. Therefore, if the paper can present reasonable and convincing results, it could become a high-impact piece of work.

**Weaknesses:**

However, I feel that this paper fails to convincingly argue why the FW loss function should replace cross-entropy.

The main justification given for using the FW loss instead of cross-entropy is that it facilitates the implementation of privacy-preserving AI under fully homomorphic encryption (FHE). The argument is that transcendental functions (e.g., logarithms) are difficult to compute exactly under FHE, so FW loss — which does not involve such functions — would be more suitable than cross-entropy.

As someone who has conducted continuous research in FHE-based privacy-preserving AI, I find this reasoning difficult to accept. This is because the proposed approach relies on max-heap operations for efficiency. In fact, comparison operations are among the most challenging computations in FHE — even more so than many transcendental functions. The max-heap data structure inherently requires frequent comparisons whenever data is inserted or removed, resulting in significant computational overhead when implemented homomorphically.

In contrast, transcendental functions can often be approximated quite effectively with polynomial approximations — as long as the interval is well chosen and the polynomial degree is sufficiently high — so the computational overhead is not as severe as one might expect. While replacing transcendental functions with polynomial-only loss functions is certainly desirable from the perspective of FHE-based AI research, I believe that using the current FW loss function in an FHE setting remains highly impractical.

To convincingly demonstrate the practicality of FW loss for FHE-based AI, the authors would need to actually implement the algorithm under FHE and show that the runtime is significantly reduced compared to cross-entropy. Without such an experiment, the claimed efficiency under FHE cannot be substantiated. From the current paper, it appears that there is no clear reason to use FW loss instead of cross-entropy outside the privacy-preserving AI context.

Moreover, the dataset used to show that FW loss can replace cross-entropy seems **too small**. To make a compelling case for substitution, experiments on larger and more complex datasets would be necessary.

**Questions:**

1. If the FW function is to be utilized under FHE, can you describe how the algorithm would need to be modified using FHE operations?
2. Simply using transcendental functions is not inherently problematic in FHE as long as they can be reasonably approximated by polynomials of appropriate degree. Have you examined how high the polynomial degree must be when using cross-entropy? If so, it would be helpful to describe that in the paper.

---

> ### Author Response · Authors · 2025-11-19
>
> We deeply appreciate the feedback from the reviewer, especially since the reviewer is an expert in FHE.
>
> In current FHE schemes (e.g., CKKS), the FW loss would indeed be quite expensive to implement due to the comparisons. However, existing FHE schemes are designed to support implementing low-degree polynomials. It is possible that a (possibly new) FHE scheme that go a different route would be better than losses like FW loss.
>
> Our contributions in this work are 1. theory that supports efficient computation of the gradient of FW loss and 2. demonstrate that hyper-hyperoptimizers can give advantage to the FW loss. Hence, regardless of whether such (possibly new) FHE scheme exist, we believe our work has merit. It is difficult to establish advantage of a new loss over the incumbent cross entropy from every aspects. In our work, we focused on the two aspects aforementioned, and hope our work to inspire new FHE schemes as a potential direction of future work.

---

### Meta-Review · Area_Chair_X3n5 · 2026-01-06

**Summary:**

In a nutshell, this paper studies the familywise (FW) loss as an alternative to cross-entropy (CE) for multiclass classification, motivated in part by the desire to avoid transcendental functions (exp/log) that can be problematic in certain computational regimes (e.g. FHE). The central technical contribution is a set of theoretical results enabling efficient FW gradient computation using retrieval-style algorithms (top‑k / selection rather than full sort), including a bitonic/unimodality property used to justify faster-than-sorting approaches, plus implementations targeting multicore CPU and a pure PyTorch GPU path.

Across reviews, there is agreement that the algorithmic/theoretical angle is interesting and that the paper is generally well-written, with useful proof sketches and practical implementation details.
However, there is limited consensus on (i) whether the paper’s motivation (especially FHE) is adequately supported, and (ii) whether the empirical evaluation is sufficiently strong and representative to justify acceptance at ICLR.

Two reviewers were marginally positive, highlighting the novelty of connecting FW gradients to retrieval-style computation and appreciating the implementations and experimental evidence in the parameter-free optimiser regime.

One reviewer was marginally negative, acknowledging the theory/engineering but emphasising missing experiments (privacy-preserving setting), limited scale, and lack of results with standard optimisers. After rebuttal, they explicitly kept their score and reiterated these concerns.

One reviewer was strongly negative, arguing that the FHE motivation is unconvincing because the proposed efficiency hinges on comparison-heavy structures (heaps), which are themselves hard under FHE; they also found the datasets too small and asked for larger-scale evidence or an actual FHE implementation/benchmark.

**Reviewer Concerns:**

The main overarching concern here is the weak author responses, which leave most concerns partially addressed or not addressed at all.

For instance:

a) Clarification of the CIFAR‑100 failure mode/stability issue: Reviewer GTVe questioned numerical stability and applicability to large-class settings. The authors clarified that the cited failure was tied to GDTUO behaviour and occurred for both CE and FW, suggesting an optimiser issue/bug rather than FW-specific instability; they also clarified their intent in choosing hyper-hyperoptimisers for a “blank slate” comparison.

b) FHE / privacy-preserving efficiency claims lack evidence...Both Reviewer VjWb and Reviewer GTVe requested substantiation of the FHE motivation, ideally an implementation/benchmark or at least a more credible accounting of costs. VjWb argued comparisons (needed for heaps/top‑k logic) are particularly challenging under FHE and that polynomial approximations often handle exp/log reasonably, so transcendental-free alone is not a decisive advantage.
In rebuttal, the authors agreed that, under current schemes (e.g. CKKS), FW would be expensive due to comparisons and that the work should be viewed more as inspiration for possible future FHE schemes.

c) Missing results with common optimisers / tuned baselines...A key point for Reviewer GTVe is that results are shown primarily with GDTUO and DoG (and Prodigy for linear models), and they are not convinced why standard optimisers (e.g. SGD/Adam variants) tuned for FW are not presented; they explicitly maintain their score because this remains unaddressed.

**Reviewer Scores:**

Given the author responses and initial reviews, I believe scores would have remained unchanged across the board.

---

### Decision · Program_Chairs · 2026-01-26

Reject